# A Collaborative Model to Implement Flexible, Accessible and Efficient Oncogenetic Services for Hereditary Breast and Ovarian Cancer: The C-MOnGene Study

**DOI:** 10.3390/cancers13112729

**Published:** 2021-05-31

**Authors:** Julie Lapointe, Michel Dorval, Jocelyne Chiquette, Yann Joly, Jason Robert Guertin, Maude Laberge, Jean Gekas, Johanne Hébert, Marie-Pascale Pomey, Tania Cruz-Marino, Omar Touhami, Arnaud Blanchet Saint-Pierre, Sylvain Gagnon, Karine Bouchard, Josée Rhéaume, Karine Boisvert, Claire Brousseau, Lysanne Castonguay, Sylvain Fortier, Isabelle Gosselin, Philippe Lachapelle, Sabrina Lavoie, Brigitte Poirier, Marie-Claude Renaud, Maria-Gabriela Ruizmangas, Alexandra Sebastianelli, Stéphane Roy, Madeleine Côté, Marie-Michelle Racine, Marie-Claude Roy, Nathalie Côté, Carmen Brisson, Nelson Charette, Valérie Faucher, Josianne Leblanc, Marie-Ève Dubeau, Marie Plante, Christine Desbiens, Martin Beaumont, Jacques Simard, Hermann Nabi

**Affiliations:** 1Centre de Recherche du CHU de Québec-Université Laval, Hôpital du Saint-Sacrement, 1050, Chemin Ste-Foy, Local J0-01, Québec, QC G1S 4L8, Canada; Julie.Lapointe@crchudequebec.ulaval.ca (J.L.); Michel.Dorval@pha.ulaval.ca (M.D.); jocelyne.chiquette@chudequebec.ca (J.C.); jason.guertin@fmed.ulaval.ca (J.R.G.); maude.laberge@fsa.ulaval.ca (M.L.); karine.bouchard@crchudequebec.ulaval.ca (K.B.); jacques.simard@crchudequebec.ulaval.ca (J.S.); 2Centre de Recherche CISSS Chaudière-Appalaches, 143 Rue Wolfe, Lévis, QC G6V 3Z1, Canada; johanne_hebert@uqar.ca; 3Faculté de Pharmacie, Université Laval, 1050 Av de la Médecine, Québec, QC G1V 0A6, Canada; 4CHU de Québec-Université Laval, 1050, Chemin Ste-Foy, Québec, QC G1S 4L8, Canada; jean.gekas@chudequebec.ca (J.G.); josee.rheaume@chudequebec.ca (J.R.); karine.boisvert@chudequebec.ca (K.B.); claire.brousseau@chudequebec.ca (C.B.); lysanne.castonguay@chudequebec.ca (L.C.); sylvain.fortier@chudequebec.ca (S.F.); isabelle.gosselin@chuq.qc.ca (I.G.); philippe.lachapelle@chudequebec.ca (P.L.); sabrina.lavoie@chudequebec.ca (S.L.); brigitte.poirier.med@ssss.gouv.qc.ca (B.P.); marie-claude.renaud.med@ssss.gouv.qc.ca (M.-C.R.); maria-gabriela.ruizmangas@chudequebec.ca (M.-G.R.); alexandra.sebastianelli@fmed.ulaval.ca (A.S.); stephane.roy@chudequebec.ca (S.R.); parole.onco@chudequebec.ca (M.C.); Marie.Plante@crchudequebec.ulaval.ca (M.P.); christinedesbiens@live.ca (C.D.); martin.beaumont@chudequebec.ca (M.B.); 5Institut de Recherche du Centre Universitaire de Santé McGill, 2155 Rue Guy, 5e étage, Montréal, QC H3H 2R9, Canada; Yann.Joly@mcgill.ca; 6Département de Génétique Humaine et Unité de Bioéthique, Faculté de Médecine, Université McGill, 3605 Rue de la Montagne Montréal, Montréal, QC H3G 2M1, Canada; 7Département de Médecine Sociale et Préventive, Faculté de Médecine, Université Laval, 1050 Avenue de la Médecine, Université Laval, Québec, QC G1V 0A6, Canada; 8Vitam, Centre de Recherche en Santé Durable, Université Laval, 2525, Chemin de la Canardière, Québec, QC G1J 0A4, Canada; 9Département des Opérations et Systèmes de Décision, Faculté des Sciences de l’Administration, Université Laval, 2325 Rue de la Terrasse Université Laval, Québec, QC G1V 0A6, Canada; 10Département des Sciences Infirmières, Université du Québec à Rimouski (UQAR), Campus de Lévis, 1595 Boulevard Alphonse-Desjardins, Lévis, QC G6V 0A6, Canada; 11Centre de Recherche du CHUM, 900, Rue Saint-Denis, Montréal, QC H2X 0A9, Canada; marie-pascale.pomey@umontreal.ca; 12Département de Gestion, Évaluation et Politique de Santé, Faculté de Médecine, Université de Montréal, 7101 Avenue du Parc, 3e Étage, Montréal, QC H3N 1X9, Canada; 13CIUSSS Saguenay Lac-St-Jean, 930 Rue Jacques-Cartier Est, Chicoutimi, QC G7H 7K9, Canada; tania.cruzmarino.csssc@ssss.gouv.qc.ca (T.C.-M.); omar.touhami.med@ssss.gouv.qc.ca (O.T.); sylvain.gagnon.med1@ssss.gouv.qc.ca (S.G.); valerie.faucher@ssss.gouv.qc.ca (V.F.); josianne.leblanc@ssss.gouv.qc.ca (J.L.); marie.eve.dubeau@ssss.gouv.qc.ca (M.-È.D.); 14CISSS Bas St-Laurent, 150 Av Rouleau, Rimouski, QC G5L 5T1, Canada; arnaud.blanchet-saint-pierre.1@ulaval.ca (A.B.S.-P.); marie-claude.roy.cisssbsl@ssss.gouv.qc.ca (M.-C.R.); nathalie.cote0104.cisssbsl@ssss.gouv.qc.ca (N.C.); carmybrisson@hotmail.fr (C.B.); nelson.charette.cisssbsl@ssss.gouv.qc.ca (N.C.); 15CISSS Chaudière-Appalaches, 101 Rue du Mont-Marie, Lévis, QC G6V 5C2, Canada; Marie-MichelleRacine@ssss.gouv.qc.ca; 16Département de Médecine moléculaire, Faculté de Médecine, Université Laval, 1050 Avenue de la Médecine, Québec, QC G1V 0A6, Canada

**Keywords:** breast and ovarian cancer, health services delivery, integrated knowledge translation, interdisciplinary research, oncogenetics, prevention, screening, treatment

## Abstract

**Simple Summary:**

We recently developed an oncogenetic model to overcome the unprecedented demand for genetic counseling and testing for hereditary breast and ovarian cancer. Quality and performance indicators showed that the implementation of this model improved access to genetic counseling and minimized delays for genetic tests for patients, who reported to be overwhelmingly satisfied with the process. However, it remains unknown whether this model is robust and sustainable or requires adjustments. In addition, whether the model could be deployed elsewhere remains also to be elucidated. The C-MOnGene study was therefore designed to gain an in-depth understanding of the context in which the model was developed and implemented, and document the lessons that can be learned to optimize oncogenetic services delivery in other settings.

**Abstract:**

Medical genetic services are facing an unprecedented demand for counseling and testing for hereditary breast and ovarian cancer (HBOC) in a context of limited resources. To help resolve this issue, a collaborative oncogenetic model was recently developed and implemented at the CHU de Québec-Université Laval; Quebec; Canada. Here, we present the protocol of the C-MOnGene (Collaborative Model in OncoGenetics) study, funded to examine the context in which the model was implemented and document the lessons that can be learned to optimize the delivery of oncogenetic services. Within three years of implementation, the model allowed researchers to double the annual number of patients seen in genetic counseling. The average number of days between genetic counseling and disclosure of test results significantly decreased. Group counseling sessions improved participants’ understanding of breast cancer risk and increased knowledge of breast cancer and genetics and a large majority of them reported to be overwhelmingly satisfied with the process. These quality and performance indicators suggest this oncogenetic model offers a flexible, patient-centered and efficient genetic counseling and testing for HBOC. By identifying the critical facilitating factors and barriers, our study will provide an evidence base for organizations interested in transitioning to an oncogenetic model integrated into oncology care; including teams that are not specialized but are trained in genetics.

## 1. Introduction

It has been estimated that highly penetrant hereditary cancer syndromes account for up to 10% of all cancer incidence worldwide [1]. While these cancers represent a small fraction of the overall cancer burden [2], they tend to develop earlier in life and to be more aggressive than so-called “sporadic” cancers [3,4]. The most common hereditary cancer syndromes are hereditary breast-ovarian cancer (HBOC) and colorectal cancer (CRC) syndromes. About 5% to 10% of breast cancer cases are estimated to be directly attributable to inherited pathogenic genetic variants in high penetrance susceptibility genes passed on through generations [5,6]. Pathogenic variants in the *BRCA1* or *BRCA2* genes are the most common cause of HBOC, and variant carriers have a 5- to 20-fold increased risk of developing breast or ovarian cancer [7,8]. The penetrance or cumulative lifetime risk of developing breast cancer is estimated at 72% for *BRCA1* and 69% for *BRCA2* pathogenic variant carriers [9]. The corresponding figures for ovarian cancer are 44% and 17% [9].

Knowledge of *BRCA* pathogenic variant status can inform surgical management for breast cancer patients, with more radical surgery considered to be an effective treatment in variant carriers, who are at increased risk of subsequent cancer [10]. This information might also change chemotherapy treatment decisions, given that some treatments are more effective than others in *BRCA* pathogenic variant carriers. Indeed, although platinum-based therapies are not the standard of care for breast cancer, they can have utility in carriers of pathogenic mutations in *BRCA1* and *BRCA2* genes, who are particularly sensitive to platinum-based drugs [11]. Newer effective personalized therapies, including poly-ADP-ribose polymerase (PARP) inhibitors targeting *BRCA1* or *BRCA2* pathogenic mutations directly or via their pathways are increasingly available for ovarian and breast cancer patients who are carriers of such variants [11].

The value of *BRCA1* and *BRCA2* testing for cancer risk reduction in breast and ovarian cancer is well-established [12,13,14]. Compelling evidence has shown that several interventions are effective in reducing the risk of future breast and ovarian cancer of women with *BRCA1* or *BRCA2* pathogenic variants. Bilateral mastectomy has been shown to decrease the risk of breast cancer by approximately 90% [15,16]. For risk-reducing bilateral salpingo-oophorectomy, the risk for ovarian cancer has been shown to decrease by 80–96% [17,18,19]. Chemopreventive interventions, including the use of oral contraceptive pills for the prevention of ovarian cancer and the use of tamoxifen for the prevention of breast cancer, have also been shown to be effective strategies [20,21]. Adding magnetic imaging resonance (MRI) to screening was found to be cost-effective for *BRCA1* pathogenic variant carriers [22].

Recognizing that the identification of individuals at risk of carrying a *BRCA1* or *BRCA2* pathogenic variant allows for targeted screening, prevention, and treatment, and can be lifesaving, several international guidelines recommend genetic testing for these genes within the set of possible genetic variants for individuals or patients suspected of being at risk of breast and ovarian cancer [12,13,23]. In the province of Quebec, Canada, after rigorous evaluation of the evidence, the National Institute of Excellence in Health and Social Services (INESSS) recommended that the Ministry of Health and Social Services (MSSS) include genetic testing for the detection of *BRCA1* and *BRCA2* pathogenic variants for breast and ovarian cancer in the medical biology procedures directory [24]. In theory, when these tests are indicated, they are available and covered by Quebec’s healthcare plan. However, in practice access to these tests remains a major challenge for patients and their families, health professionals, and the healthcare system due to a demand that exceeds capacity.

Medical genetic services in the province of Quebec, as in many other Canadian jurisdictions and countries, are facing an unprecedented demand for genetic counseling (GC) and genetic testing (GT) for HBOC, a demand that exceeds the systems’ current capacity [25,26]. Of significance is the fact that access, particularly wait times to be seen in genetic counseling services, is not equitable across Canada. Across its 10 provinces and 3 territories, the ratio of genetic counselors in Canada vary between 1 per 100,000 people and 1 per 253,000 people [27]. These estimates date back to 2013, but given the limited number of students graduating from Canadian genetic counseling programs [28], they are likely still valid. The Quebec Association of Geneticists estimates that there are only 33 physicians specialized in medical genetics in the province, for a population of 8.5 million, and not all are working in oncology [29]. Consequently, it is difficult for the staff to meet the ever-growing demand for testing, particularly in a context where expedited multi-gene cancer panel testing is now ordered as a first-line approach [30]. This situation results in long waiting lists and delays, and patient dissatisfaction, as wait times can vary from a few days to several years, depending on priority level [31].

One could say that an easy solution might be to recruit more staff, particularly genetic counselors. However, in Canada, there are five graduate programs for genetic counselors, which cumulatively generate approximately 20–25 new genetic counselors per year. Among these students, 4–6 are from the province of Quebec [28], and 50% of them are expected to focus on cancer. This strategy is therefore unlikely, in the short and medium terms, to address the shortage of oncogenetic resources we are witnessing in Canada [31,32] and other countries [33].

In light of increasing imbalance between the supply and demand for genetic counseling and testing for HBOC, alternative models of genetic service delivery have been called for [34,35,36,37]. Innovative models developing in local and regional settings have the potential to optimize the efficiency and uptake of existing testing opportunities. One of such models is the Collaborative Oncogenetic Model (COM) developed by the CHU de Québec-Université Laval, a model which differs from the traditional genetic information delivery model and innovates on several aspects (see Figure 1). First, the oncogenetic team is interdisciplinary and integrated throughout the care trajectory. It prioritizes interprofessional collaboration by extending the role of health professionals not specialized in genetics in the context of care. Thus, several clinicians (oncologist surgeons, gynecologic oncologists, general practitioners) can order genetic tests, in collaboration with geneticists from the medical genetics service. Nurse navigators and clinical nurses in oncology trained in oncogenetics, working in collaboration with genetic counselors, identify people likely to benefit from a genetic test, obtain their informed consent, and provide genetic counseling and order testing. Second, the model offers diverse and personalized genetic services. During pre-test counseling, genetic group counseling (i.e., 15 to 25 people) is favored over individual counseling. The disclosure of the results (post-test counseling) is personalized according to results, clinical context, and patients’ place of residence and is reported by letter, phone, or in person. Patient partners are integrated into the team and actively participate in both organizational and clinical activities. Third, the model also prioritizes inter-institutional collaboration and shared responsibility between the different health institutions taking part in an integrated university health network. This collaboration makes it possible to better equip primary and secondary health care centers of the network so that they can contribute to meet the needs of patients likely to be referred to oncogenetics. With regard to the goal, the COM resembles the mainstream cancer genetics (MCG) program developed and implemented in the United Kingdom (UK) [38,39]. However, in contrast to the MCG, the COM activities target the entire trajectory, from counseling to following up with patients in an integrated fashion. Another notable difference is the fact that MCG only concerns cancer patients whereas the COM targets also unaffected patients.

After a preparatory phase between 2015 and 2016, the COM was implemented in 2017, and has been continuously updated, when needed, to preserve its flexibility. Globally, the model has improved the delivery of oncogenetic services to breast and ovarian cancer patients and high-risk healthy women through a reorganization of existing resources and expertise. As shown in Figure 2, the total number of genetic consultations significantly increased in 2018 and more than doubled in 2019 when compared to 2016. By extending the role of health professionals not specialized in genetics and by adding group counseling sessions, more than 1640 individuals on the waiting list were able to benefit from genetic counseling services. Among these, 370 breast and ovarian cancer patients awaiting a surgical or therapeutic decision had direct access to the genetic test through their oncologist. Improved access to oncogenetic services was also observed for patients from remote areas. More than 350 patients from a peripheral region were able to access a genetic test or an appropriate triage mechanism.

In addition, as shown in Figure 3, the average number of days between genetic counseling and disclosure of test results significantly decreased after the model implementation in 2017, with a reduction of almost 100 days between 2016 and 2019—a decrease that might also be partly explained by improvement in turnaround time required by genetic testing labs to provide results. Furthermore, a survey conducted among patients showed a very good level of satisfaction. For instance, regarding group counseling sessions, patients felt that they were well informed and supported. Notably, as presented in Figure 4, 100% of patients (N = 870) agreed or totally agreed that “the content addressed was appropriate and comprehensive” or that the “nurses used clear and simple words to communicate with me”. A vast majority agreed or totally agreed that the information provided was useful (98%), and they understood it (99%). Interestingly, 96% of participants agreed or totally agreed that they were comfortable expressing themselves in the group, suggesting that these group counseling sessions seem to have empowered patients and improved their perception of control, which allowed them to make an informed decision regarding genetic testing.

Finally, these data also showed that group counseling sessions improved participants’ understanding of breast cancer risk and increased knowledge of breast cancer and genetics. As shown in Figure 5, the percentage of correct answers significantly improved for all items after the counseling (*p* values of all paired *t*-tests < 0.001), which is consistent with patients’ satisfaction with the process.

The COM has been developed and implemented through a natural emergent mechanism [40] stemming from a clinical team’s desire to respond to the population needs in a context of scarce resources and efficiently integrate evidence-based knowledge from genetics and genomics into the care trajectory. Based on data mentioned above, it is reasonable to say that this collaborative model seems to offer more flexible, patient-centered, and efficient genetic counseling and testing for cancer patients, who reported to be overwhelmingly satisfied with the process. However, it is unknown whether the model is robust and sustainable or requires adjustments. In addition, whether the activities of the model could be deployed elsewhere and at what conditions remains to be elucidated.

The overarching aim of C-MOnGene study is to gain an in-depth understanding of the context in which the COM was developed and implemented, and document the lessons that can be learned from it to optimize local and regional oncogenetic services delivery in Quebec and elsewhere. Our hypothesis is that a dynamic and complex interplay of contextual factors (e.g., medical guidelines and facilities, budget, regulation/norms), actors (e.g., patients/families, healthcare providers and managers, health authorities, companies) and culture (e.g., value of care, therapeutic paradigms, priority setting) influenced the development and implementation of the COM. We also hypothesize that each site involved in delivering oncogenetic services will experiment its unique interplay of factors. Our research will be guided by the following specific questions (RQ):

RQ1: What general and specific factors influenced the development and implementation phases of the COM, and what adjustments are still required to further enhance its performance?

RQ2: Which components of the COM could be adopted, in what types of settings, and what are the critical barriers and facilitators?

RQ3: What are the social (acceptability), economic, legal, and regulatory implications of implementing the COM?

RQ4: What can be derived for future planning and implementation of new (or adapted) genetic health services in oncology settings?

## 2. Research Methods and Analyses

Our hypothesis is that a dynamic and complex interplay of contextual factors (e.g., medical guidelines and facilities, budget, regulation/norms), actors (e.g., patients/families, healthcare providers and managers, health authorities, companies) and culture (e.g., value of care, therapeutic paradigms, priority setting) influenced the development and implementation of the COM. We also hypothesize that each site involved in delivering oncogenetic services will experiment its unique interplay of factors.

### 2.1. Study Design

Our research will use a multiple case study design [40] with mixed methods integrating both qualitative and quantitative data. Since the main objective of the project is to capture the processes involved in the development of the COM pathway in the real-life context of care, and document the extent to which this model could be deployed in different settings, the case study design is appropriate.

To guide our methodological choices, including data collection and analytical approaches, we will rely on two complementary conceptual frameworks to answer our research questions. The first is the diffusion of innovations theory developed by Rogers [41,42]. This theory posits that an innovation (here the COM) will be successfully adopted if it is perceived to be advantageous (relative advantage); compatible with organizational and individual values, norms, and needs (compatibility); simple to implement (complexity); usable in preliminary fashion on a small scale (trialability); clearly beneficial (observability); and adaptable (reinvention). The second theoretical model is the framework for evaluating the socio-political implications of technological innovations by Lehoux and Blume [43], which allows a contextual analysis of the evaluation of technological, organizational, or therapeutic innovations. According to this model, the social and political implications of health innovations should be studied as a dynamic two-way process rather than solely on a one-way fashion. There would be no technological innovation without socio-political properties, because its existence and its adoption require the support of social groups investing financial and human resources while engaging in relations of power. The model allows accounting for potential interactions between innovation and: (1) actors (promoters, health professionals, managers, patients, public, media, decision-makers); (2) knowledge (evidence available or to be produced); (3) resources (material, human, and financial), and (4) power plays (allocation of resources, political interests, preferences).

### 2.2. Cases Selection

In agreement with the research questions and the objective of examining the potential adoption of the COM components in a variety of settings with distinct clinical missions, population structures and needs, medical facilities and healthcare human resources, our research activities will be conducted in four settings. Setting 1 is where the COM was initially developed and has run successfully since 2017. This setting is part of a network of five University-affiliated hospitals, which constitutes one of the three largest university hospital centers in Canada deserving nearly two million people for a land area of 18,600 km^2^. The three other settings (Settings 2, 3 and 4) are health and social services integrated centers called CISSS and CIUSSS (depending on university affiliations) which include hospitals, primary care and community services. These three settings deserve populations varying between 197,000 people and 428,000 people for land areas varying between 15,000 and 95,700 km^2^.

### 2.3. Data Collection and Procedures

RQ1: What general and specific factors influenced the development and implementation phases of the COM, and what adjustments are still required to further enhance its performance?

To identify the events and processes that led to the development and implementation of the COM at Setting 1, we will start by reviewing pertinent documents, including meeting minutes, action plans, internal notes, and annual reports. A series of one-to-one semi-structured interviews will be conducted with key informants, including members of the interdisciplinary team who initiated the model, medical directors of the breast cancer clinic and gynecologic oncologic division, and front-line managers of the oncology and genetics clinical divisions. Purposive sampling will be used to identify these informants, with snowball sampling used to further include individuals involved in the COM. The interviews will explore external and internal factors underlying the development of this model. We will also explore experiences, barriers and challenges related to its development and the conditions in which the activities have been delivered (staff’s motivations, perspectives, organizational infrastructures and logistics, resources and policies). Discussions will also concern the sustainability of these activities, potential areas of concern, and the need (or not) for adjustments. We will also seek patients’ perspectives regarding the model. Patients who consented to group counseling and those who accepted the genetic test since the model’s inception will be asked to fill the validated French version of the decision regret scale (5 items) [44]. They will also be asked to complete a short questionnaire related to satisfaction with the process of genetic testing and will be invited to provide free text recommendations to improve the process of the model [45]. Lastly, a deliberative focus group discussion, conducted by a professional moderator, will be organized with all stakeholders to further explore and validate the various aspects raised through documentary analyses, interviews and responses to questionnaires. This entire process will allow drawing a clear portrait of the organizational, logistics, and resource requirements to implement one or several components of the COM pathway.

RQ2: Which components of the COM could be adopted, in what types of settings, and what are the critical barriers and facilitators?

To address this question, we will first undertake an environmental scan of all four sites to assess the current state of their clinical cancer genetic services. Thus, demographics, number of patient referrals to general and more specific cancer genetic resources, number of patients seen over the last 12 months, dedicated genetics resources, referrals and scheduling processes, resources and funding, and documented policy for identifying and managing urgent cases will be extracted through medical charts (databases and/or spreadsheets). Responses to a dedicated questionnaire addressed to genetics departments or health professionals dealing and/or overseeing genetics services delivery will also serve to that purpose. Second, an open-ended questionnaire accompanied by a summary of key elements of the COM (process, required resources and impacts) will be addressed to medical oncologists, geneticists and genetic counselors, nurse navigator and clinical nurses, and front-line managers of Settings 2, 3, and 4 to gather their opinions, beliefs, and feelings about the COM. They will also be questioned on whether the model or some of its components could be implemented in (or adapted to) their setting in addition to related challenges, opportunities, threats, and mitigation strategies. Specific important issues that will need to be addressed concern the involvement of non-genetic health professionals in the provision of genetic services, the collaboration with geneticists and genetic counselors, and training issues. A gap analysis between the model requirements and the current situation at each site will be conducted. Using a gap analysis template, the analysis will focus on identifying the difference between current state (e.g., available resources) and the future state (e.g., required resources). Focused discussions will be organized at each site to present the results for validation and consolidation and to collaboratively define next steps and proposals.

RQ3: What are the social, economic, legal, and regulatory implications of implementing the COM?

Acceptability (concerns and barriers) from the professionals’ perspective about an extension of the role of non-genetic health professionals will be examined through focused discussions with genetics and associated health care providers from all of the four settings. Our discussion grid will be developed based on responses obtained from the open-ended questionnaire (RQ2). Acceptability (concerns and barriers) from the patients’ perspective collected at Setting 1 will be assessed by comparing the number of those who, for example, attended group counseling to those who attended individual counseling and comparing their level of satisfaction, decision regret and, when available, their usage of screening and risk-reduction strategies. A scoping review [46] will be carried out to identify legal, regulatory, and ethical issues related to the extension of the role of non-genetic health professionals and the establishment of a registry of patients who benefit from genetic counseling and testing for HBOC. A comparative law analysis will be used to identify key legal variations and promote better coordination of legal texts from different national systems. The method consists of a systematic study of legal traditions and rules on a comparative basis. A modified version of the micro-comparison approach (thematic comparative study) will be used to catalogue and study existing standard solutions in Quebec, other Canadian provinces and countries such as the UK. The applicable medico-legal and regulatory frameworks will be investigated through discussions with key stakeholders (medical law experts, hospital legal experts, clinical ethicists) and document reviews (essays, articles, case law, laws including bills). An analysis will also be done regarding health regulations bearing on the role of non-genetic healthcare professionals through the provision of genetic health services to see whether they need to be amended to allow these professionals to take an extended role. Finally, we will explore with our stakeholders and through our document reviews the feasibility and considerations of putting in place a provincial registry of patients who benefitted from genetic counseling or testing for HBOC in order to facilitate continuing service improvement initiatives. For the economic evaluation, the first step will be to collect adequate data on the resources used for the implementation of the model and assign appropriate costs [47]. Specific cost components, assumptions, and sources of unit costs will be retrospectively determined through documentary analysis and interviews with stakeholders including health care professionals, managers, financial and clinical performance officers and patients. Resource use associated with the implementation of the COM will then be collected and organized using a customized cost data capture template. Categories of cost will include direct labor, indirect labor, and non-labor costs.

RQ4: What can be derived for future planning and implementation of new (or adapted) genetic health services?

Lessons learned from this research, particularly in terms of recommendations to improve genetic and oncogenetic service delivery, will be operationalized through an invitational stakeholders’ workshop and the instigation of a clinical advisory committee (CAC) who will lead the validation of recommendations. Members of our research team organize a biannual symposium on hereditary cancer risk titled “Symposium Risque Héréditaire de Cancer,” and it is attended by more than 200 researchers, health professionals, decision makers, and patients. The workshop will be held a day before the next symposium planned in October 2021 or later depending of the situation related to the COVID-19 pandemic. The workshop participants will be determined by the research team and will include a diverse group of researchers, healthcare providers, managers, decision-makers, patient partners, and trainees with various profiles. Before the workshop, syntheses of relevant literature on the integration of innovation and collaborative models for service delivery, drawing on the methodological approach by McMaster Health Forum [48], will be provided to attendees. The workshop day will have two phases. The first phase will consist of presenting the concept, process, and results of the COM, followed by the presentation of the findings related to research questions 1 to 3. Two external experts from regions where recent strategies were developed to improve genetic testing service delivery will be invited to share their experiences. The second phase will include four focused workshop discussions: strategies to improve oncogenetic health service delivery, particularly for breast and ovarian cancer; realistic scenarios to implement all or some components of the COM; education and training of nongenetic health professionals; and standardized criteria for genetic referrals and testing, in particular for HBOC. Attendees will be divided into four-balanced groups to discuss each issue. Attendees will then be reassembled to present key elements of their discussions and their priority recommendations to the whole group for further discussions. All presentations and discussions will be documented and summarized using field notes and audio recordings. Summaries will be further validated by attendees and sent to the members of the CAC, who will oversee the rigorous process of issuing key recommendations. Finally, the CAC will propose conditions to put in place for their optimal implementation and will set the basis for the development of an implementation tool.

### 2.4. Data Management and Analysis

All qualitative data will be retained as written or electronic files, including document summaries, open-ended questionnaire responses, and interviews and group discussion transcripts. Interviews and group discussions will be recorded and transcribed verbatim. Thematic analysis will be used to systematically analyze all qualitative data [49] using NVivo software. All interview data will be coded to protect the confidentiality of participants’ information. Furthermore, special care will be taken to ensure that comments cannot be attributed to a single individual. Quantitative data from surveys, questionnaires, and administrative data will be analyzed using SAS software, Version 9.4 of the SAS software for Windows (Copyright © 2021 by SAS Institute Inc., Cary, NC, USA) and descriptive and inferential statistics will be provided. More detailed modeling of variations between participants or sites will be considered if our data permits. We will report any missing data from our surveys and describe the potential impact, if any, in our findings. We will ensure quantitative and qualitative analyses build upon one another by cross-checking document reviews with findings from interviews, deliberative forums, and workshops (e.g., qualitative data used to explain quantitative findings or quantitative data used to test hypotheses generated by qualitative data). The cost of implementing the COM and its projected impact on patients’ health care utilization will be combined within a budgetary impact analysis. Micro-costing will be used to calculate the incremental costs of the COM compared to usual practice [47,50]. To provide a clear descriptive summary for stakeholders, particularly decision-makers, a cost-consequences analysis will be conducted. The cost-consequences analysis will provide an estimate of the incremental cost of implementing the COM alongside its outcomes (e.g., access to oncogenetic counseling and/or testing for HBOC) compared to a traditional oncogenetic model. [51,52].

## 3. Discussion

We sought here to present the protocol of the C-MOnGene study, designed to improve our understanding on how a novel collaborative oncogenetic model for genetic counseling and testing for HBOC was developed, performs, and could be integrated in different settings. We also presented quality and performance indicators that prompted us to develop this protocol. Indeed, these data suggest that implementing the model enabled clinical teams to provide more consultations and decrease the average number of days between counseling and disclosure. However, more in-depth analyses of these findings are required since historical changes have occurred and are occurring at a fast pace in the field of genetics in areas such as recommendations, access and turnaround time get results [53,54,55].

The demand for oncogenetic services has increased in recent years in the context of limited resources, resulting in dramatic increases in delays for genetic counseling and testing for high-risk healthy women and breast and ovarian cancer patients. It is recognized that failure to identify individuals with a higher risk of carrying pathogenic variants for HBOC represents missed preventive and therapeutic opportunities. Like many other jurisdictions around the world, precision medicine has been identified as one of two cross-cutting niches leading the province of Quebec’s life sciences strategy 2017–2027 [56]. Precision medicine is essentially based on the development of genetic/genomic diagnostic, predictive, and prognostic tests. Without organized and facilitated access to these tests, the expected health and economic potential of this emerging medical approach will not be realized. With the dropping cost of genetic testing, media coverage of advances in genomics in cancer prevention strategies, and the personalization of treatments of several cancers including breast, ovarian, prostate, and pancreatic cancers [57], the demand for oncogenetic services will inevitably increase further [58,59]. For this reason, this research project anticipates numerous benefits. By identifying the critical facilitating factors and barriers to overcome, our results will provide an evidence base for organizations interested in transitioning to an oncogenetic model integrated into oncology care, including teams not specialized but trained in genetics. By dealing with urgent cases with clear indication, these teams will allow geneticists and genetic counselors to focus on more complex cases, which will improve access, quality, and equity in care and health services. Many patients living in rural and remote areas do not have easy and timely access to the genetic counseling services provided in urban tertiary care centers. This project will thus be of great interest to the different health system policy and decision makers as it will assess whether the COM or some of its components could be applied to other health care facilities that do not currently have access to specialized genetic services.

## 4. Ethics and Discussion

This study protocol has been examined by the Research Ethics Board of the CHU de Québec-Université Laval. The Board waived the ethics approval requirement since the study’s purpose is to evaluate the quality of services already in place at the institution.

In order to maximize the quality and impact of deliverables, our project includes an integrated knowledge translation strategy [49]. A widely recognized and accepted tenet of a successful knowledge translation strategy is the integration of knowledge users throughout the research process. We, therefore, designed this study by bringing together an outstanding collaborative team of researchers and knowledge users with diverse and complementary expertise to ensure that our findings will be robust in terms of research quality, integrity, and credibility, and will be useful for decision-makers. These team members have actively and critically informed the research questions and approaches. To raise awareness on issues related to the provision of genetic testing and oncogenetic services delivery and promote actions when we have project findings, our knowledge transfer (KT) activities will include publications in relevant peer-reviewed journals and presentations at relevant national and international conferences and workshops. We will communicate our results in institutional newsletters to inform patients, general population, and other health professionals about the study findings. We will produce a comprehensive report with policy analyses and recommendations that could be used by decision-makers when addressing critical issues raised around the development and use of genetic technologies. To help drive province-wide genetic policies, we will provide health economic, legal, and acceptability analyses evaluating the costs and the regulatory and social implications of implementing a collaborative oncogenetic model. We will also provide realistic scenarios and points to consider for institutions interested in implementing the collaborative oncogenetic model pathway or some of its components. Finally, recommendations from the clinical advisory committee (CAC) will be presented at relevant health and regulatory authorities.

## Figures and Tables

**Figure 1 cancers-13-02729-f001:**
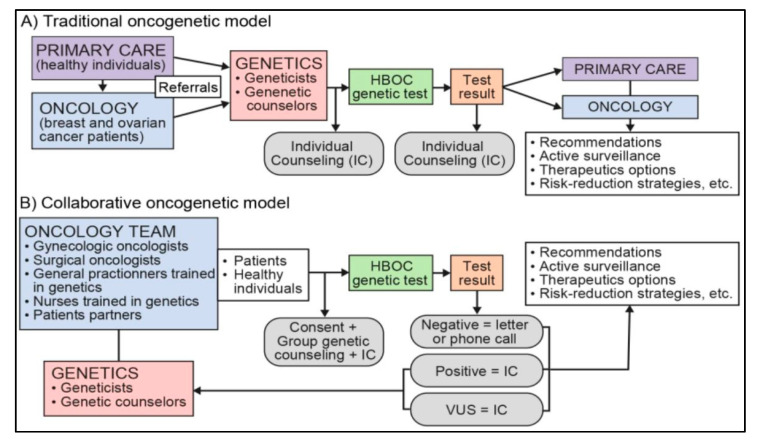
A Simplified Representation of Traditional and Collaborative Oncogenetic Models for Hereditary Breast and Ovarian Cancer (HBOC).

**Figure 2 cancers-13-02729-f002:**
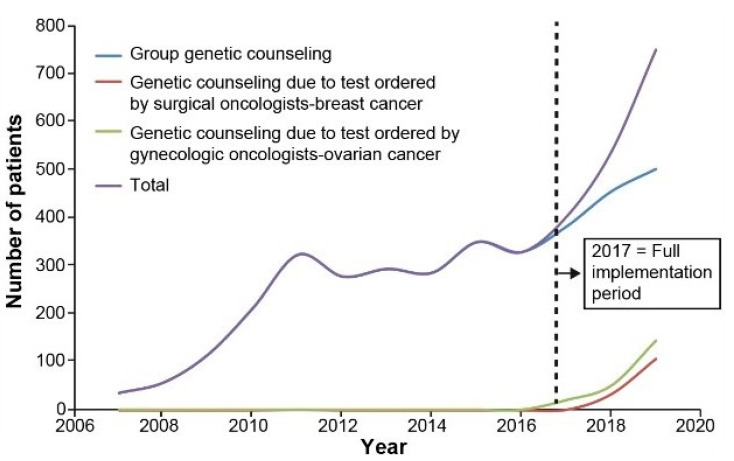
Number of Patients who Accessed Genetic Counseling for Hereditary Breast-Ovarian Cancer (HBOC) Over Time.

**Figure 3 cancers-13-02729-f003:**
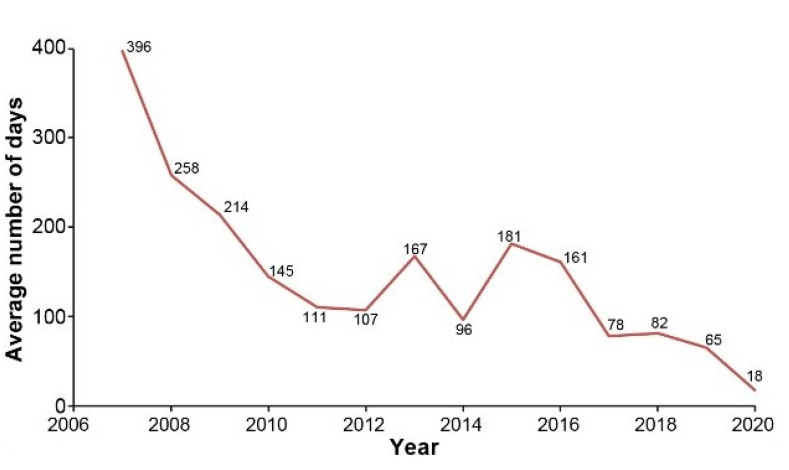
Average Number of Days between Counseling and Disclosure of Genetic Test Result Over Time.

**Figure 4 cancers-13-02729-f004:**
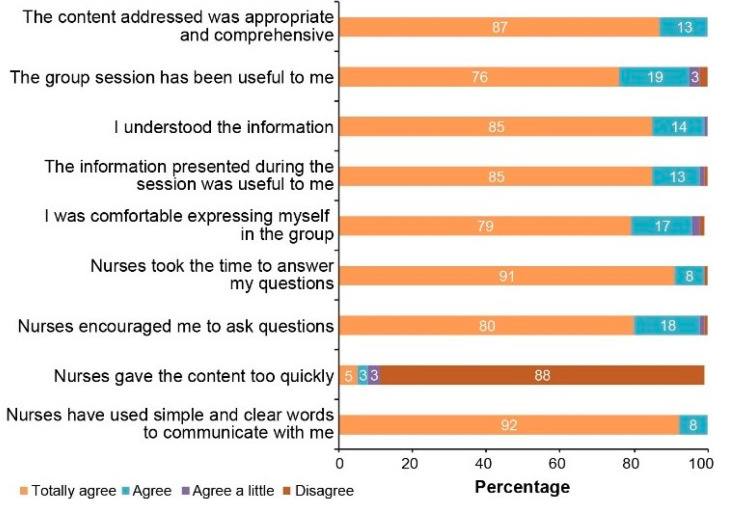
Satisfaction with Group Genetic Counseling.

**Figure 5 cancers-13-02729-f005:**
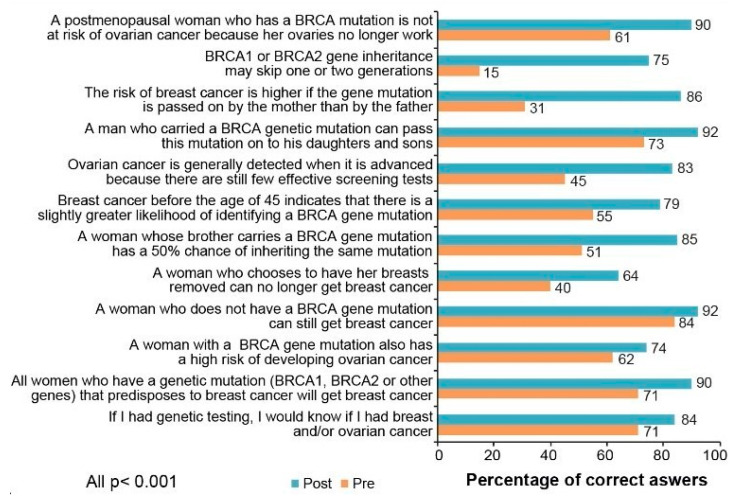
Knowledge Before and After the Counseling.

## Data Availability

Not applicable.

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
