# Peer review of "A Collaborative Model to Implement Flexible, Accessible and Efficient Oncogenetic Services for Hereditary Breast and Ovarian Cancer: The C-MOnGene Study"

_cancers, 2021, doi:10.3390/cancers13112729_

Round 1
Reviewer 1 Report
Overall this is a well considered project. What is missing from the discussion of the potential impact in the preliminary results shared is the historical changes that have occurred and are occurring over the course of time in the field of genetics, recommendations for testing, and access to testing. This may also have contributed to increases in testing. It is also not clear if changes are due to more patients being diagnosed or treated. It is clear that the intervention is impactful, it is just not clear what is the exact component making the impact and if it is due to intervention alone. It is not clear if the study protocol, with the absence of a comparison arm, will answer that question. The hypothesis as written is not clearly defined or measurable. It is also not clear how the economic impact will be measured or evaluated.
Finally, some mention of extension beyond HBOC is warranted, even if it is hypothesized.
Author Response
Response to Reviewer 1 Comments
Comment 1. Overall this is a well-considered project.
OUR RESPONSE 1: We thank the reviewer for this positive and encouraging comment.
Comment 2. What is missing from the discussion of the potential impact in the preliminary results shared is the historical changes that have occurred and are occurring over the course of time in the field of genetics, recommendations for testing, and access to testing. This may also have contributed to increases in testing. It is also not clear if changes are due to more patients being diagnosed or treated. It is clear that the intervention is impactful, it is just not clear what is the exact component making the impact and if it is due to intervention alone.
OUR RESPONSE 2: We thank the reviewer for raising this important question. We agree with him or her that the mechanisms underlying the impact of the intervention remain to be understood. This is exactly the impetus for the development of the C-MOnGene study protocol. We followed the reviewer’s recommendation and modified the discussion section as follows (page 12, lines 457-466):
“This article presents preliminary results of a collaborative oncogenetic model, recently developed and implemented at a University hospital in Quebec City, Canada, in collaboration with other regional hospitals. These preliminary results indicate that implementing the COM enabled the team to provide more consultations and decrease the average number of days between counseling and disclosure. However, more in-depth analyses of these findings are required since historical changes have occurred and are occurring at a fast pace in the field of genetics in areas such as recommendations, access and turnaround time get results [1-3]. We have thus designed the C-MOnGene study with multiple steps to improve our understanding on how this model developed, performs and could be integrated in different settings and at what conditions.”
Comment 3. It is not clear if the study protocol, with the absence of a comparison arm, will answer that question.
OUR RESPONSE 3: We agree with the reviewer that we do not have a comparison arm per se. However, this study should be seen has an implementation research which seeks to understand, within real world conditions, the general and specific factors that influenced the development and implementation phases of the collaborative oncogenetic model (COM). In addition, we are going to compare data before and after the implementation of the model at different settings, this will considerably increase the richness of our data and our ability to contextualize and cross-validate our findings.
Comment 4. The hypothesis as written is not clearly defined or measurable.
OUR RESPONSE 4: We understand the point made by the reviewer. We have clarified our hypothesis as much as we could considering that this research is focusing on the complexity of implementing innovations or changes in health care system.
Previous version: Our general hypothesis is that contextual factors (e.g. medical guidelines and facilities, budget, regulation/norms), actors (e.g. patients/families, healthcare providers and managers, health authorities, companies) and culture (e.g. value of care, therapeutic paradigms, priority setting) exert an important influence on the successful implementation (or failure) of an organizational innovation at a small scale and at its future large-scale adoption.
Revised version (pages 7, lines 245-251): Our hypothesis is that a dynamic and complex interplay of contextual factors (e.g. medical guidelines and facilities, budget, regulation/norms), actors (e.g. patients/families, healthcare providers and managers, health authorities, companies) and culture (e.g. value of care, therapeutic paradigms, priority setting) influenced the development and implementation of the COM. We also hypothesize that each site involved in delivering oncogenetic services will experiment its unique interplay of factors.
Comment 5. It is also not clear how the economic impact will be measured or evaluated.
OUR RESPONSE 5: Following this comment from the reviewer and to make it clear for readers, we provided additional information about the evaluation of the economic impact of the model as follows:
(page 10, lines 390-398):
For the economic evaluation, the first step will be to collect adequate data on the re-sources used for the implementation of the model and assign appropriate costs [47]. Specific cost components, assumptions and sources of unit costs will be retrospectively determined through documentary analysis, interviews with stakeholders including health care professionals, managers, financial and clinical performance officers and patients. Resource use associated with the implementation of the COM will then be collected and organized using a customised cost data capture template. Categories of cost will include direct labour, indirect labour, and non-labour costs.
(page 12, lines 449-455):
Micro-costing will be used to calculate the incremental costs of the COM compared to usual practice [47,50]. To provide a clear descriptive summary for stakeholders, par-ticularly decision-makers, a cost-consequences analysis will be conducted. The cost-consequences analysis will provide an estimate of the incremental cost of implement-ing the COM alongside its outcomes (e.g. access to oncogenetic counselling and/or test-ing for HBOC) compared to a traditional oncogenetic model [51,52].
Reviewer 2 Report
The article describes a collaborative prospective protocol to improve the services for genetic testing of hereditary breast and ovarian cancer.
There is a great need to develop more cost effective and more efficient protocols for genetic-based health services and in this respect I was quite eager to give the article a green light. However, when I reached the end of the article I was unsure whether I was reading a publication submission or a grant application.
Materials and methods was all in the future tense and, at first, I was looking to correct the grammar until I realised the methods were about what the authors were going to do, not done. I found the article a little confusing. On first read I was quite confused in whether there was any data, as mentioned, the protocol read like a grant proposal than an original article or a review. Then it dawned on me it was a proposal or research methodology, with preliminary data.
So I am at a crossroads in how to give a clear answer to whether this is publishable, or to ask the authors to provide some actual data.
My suggestion would be to rewrite the article, less like a proposed grant application, and more like an important step towards introducing a new methodology with substantial evidential to support a different conceptual viewpoint/alternative to provide a more efficient, informative, cost effective service, for women with genetic disorders and poor access to genetic counselling.
Therefore in its present format, I am inclined to say major revision, the decision based on the format/presentation of the article.
Author Response
Response to Reviewer 2 Comments
Comment 1. The article describes a collaborative prospective protocol to improve the services for genetic testing of hereditary breast and ovarian cancer. There is a great need to develop more cost effective and more efficient protocols for genetic-based health services and in this respect I was quite eager to give the article a green light. However, when I reached the end of the article I was unsure whether I was reading a publication submission or a grant application.
OUR RESPONSE 1: We thank the reviewer for these positive and encouraging comments.
Comment 2. However, when I reached the end of the article I was unsure whether I was reading a publication submission or a grant application. Materials and methods was all in the future tense and, at first, I was looking to correct the grammar until I realised the methods were about what the authors were going to do, not done. I found the article a little confusing. On first read I was quite confused in whether there was any data, as mentioned, the protocol read like a grant proposal than an original article or a review. Then it dawned on me it was a proposal or research methodology, with preliminary data. My suggestion would be to rewrite the article, less like a proposed grant application, and more like an important step towards introducing a new methodology with substantial evidential to support a different conceptual viewpoint/alternative to provide a more efficient, informative, cost effective service, for women with genetic disorders and poor access to genetic counselling
OUR RESPONSE 2: We thank the reviewer for this comment and we apologize for the confusion. Indeed, our manuscript presents preliminary results of the implementation of a collaborative model to improve the services for genetic testing of hereditary breast and ovarian cancer (HBOC). Although these results strongly suggest that the model improve patients’ access to genetic services for HBOC, historical changes that have occurred and are occurring over the course of time in the field of genetics, recommendations for testing, and access to testing and other factors might have had an impact. Thus, the manuscript also presents the study protocol we designed to understand, within real world conditions, the general and specific factors that influenced the development and implementation phases of the model and the conditions under which it could be implemented in other settings. In accordance with the reviewer’s comment, we made several changes to ensure our manuscript reads less like a grant proposal. We removed for example the Table 1 summarizing the research questions and associated methods. We also removed the entire paragraph detailing our knowledge transfer plan from the conclusion, as we believe it is also more in line with the style of a grant proposal. Finally, we also modify the structure of our conclusion section, which is now shorter and more straightforward. Please see pages 12-14, lines 457-544.
Round 2
Reviewer 1 Report
Thank you for your response, no further comments.
Author Response
We thank the reviewer for his/her time and consideration.
Reviewer 2 Report
Although there is some improvement in the clarity of the article, and the introduction is excellent, I still believe the article reads more like a grant proposal than an article for publication, even as a protocol/methods paper.
ie. Introduction, preliminary data, hypothesis, methods etc. I believe the style of writing of the article needs some improvement before considering for publication.
Author Response
We thank the reviewer for his/her positive and encouraging comment. We followed his/her recommendation and modified the structure of our manuscript. We believe this markedly improved the manuscript and makes it easier for readers to follow. The revised version is now focused on the presentation of the study protocol. We made sure to comply with the format specified by the Journal for this type of article (i.e. 1) Abstract, 2) Introduction, 3) Research Methods and Analysis, 4) Discussion, 5) Ethics and Dissemination). We have uploaded a clean version of the manuscript as a supplementary file in order to provide with the option of reviewing a copy of the work closer to how it will appear to readers. Again, please receive our most sincere thanks for your insights.